# Mesenchymal Stem/Stromal Cells Derived from Dental Tissues Mediate the Immunoregulation of T Cells through the Purinergic Pathway

**DOI:** 10.3390/ijms25179578

**Published:** 2024-09-04

**Authors:** Luis Ignacio Poblano-Pérez, Alberto Monroy-García, Gladis Fragoso-González, María de Lourdes Mora-García, Andrés Castell-Rodríguez, Héctor Mayani, Marco Antonio Álvarez-Pérez, Sonia Mayra Pérez-Tapia, Zaira Macías-Palacios, Luis Vallejo-Castillo, Juan José Montesinos

**Affiliations:** 1Mesenchymal Stem Cell Laboratory, Oncology Research Unit, Oncology Hospital, Centro Médico Nacional SXXI, Instituto Mexicano del Seguro Social, Mexico City 06720, Mexico; poblanoperez@hotmail.com; 2Programa de Doctorado en Ciencias Biomédicas, Universidad Nacional Autónoma de México, Mexico City 04510, Mexico; 3Immunology and Cancer Laboratory, Oncology Research Unit, Oncology Hospital, Centro Médico Nacional SXXI, Instituto Mexicano del Seguro Social, Mexico City 06720, Mexico; albertomon@yahoo.com; 4Institute of Biomedical Research, Department of Immunology, Universidad Nacional Autónoma de México, Mexico City 04510, Mexico; gladis@unam.mx; 5Immunobiology Laboratory, Cell Differentiation and Cancer Unit, Facultad de Estudios Superiores-Zaragoza, Universidad Nacional Autónoma de México, Mexico City 09230, Mexico; lulumora@yahoo.com; 6Department of Cellular and Tissue Biology, Faculty of Medicine, Universidad Nacional Autónoma de México, Mexico City 04510, Mexico; castell@unam.mx; 7Hematopoietic Stem Cell Laboratory, Oncology Research Unit, Oncology Hospital, Centro Médico Nacional SXXI, Instituto Mexicano del Seguro Social, Mexico City 06720, Mexico; hmayaniv@prodigy.net.mx; 8Tissue Bioengineering Laboratory, Postgraduate Studies, Research Division, Faculty of Dentistry, Universidad Nacional Autónoma de México, Mexico City 04510, Mexico; malvap6@gmail.com; 9Research and Development in Biotherapeutic Unit (UDIBI), National School of Biological Sciences, Instituto Politécnico Nacional, Mexico City 11340, Mexico; sperezt@ipn.mx (S.M.P.-T.); zs.macias.p@gmail.com (Z.M.-P.); lavallejos@ipn.mx (L.V.-C.); 10National Laboratory for Specialized Services of Investigation, Development and Innovation (I+D+i) for Pharma Chemicals and Biotechnological Products (LANSEIDI-FarBiotec-CONACyT), Instituto Politécnico Nacional, Mexico City 11340, Mexico; 11Department of Immunology, National School of Biological Sciences, Instituto Politécnico Nacional, Mexico City 11340, Mexico

**Keywords:** mesenchymal stem/stromal cells, adenosine, dental tissues, T lymphocytes, immunomodulation

## Abstract

Human dental tissue mesenchymal stem cells (DT-MSCs) constitute an attractive alternative to bone marrow-derived mesenchymal stem cells (BM-MSCs) for potential clinical applications because of their accessibility and anti-inflammatory capacity. We previously demonstrated that DT-MSCs from dental pulp (DP-MSCs), periodontal ligaments (PDL-MSCs), and gingival tissue (G-MSCs) show immunosuppressive effects similar to those of BM, but to date, the DT-MSC-mediated immunoregulation of T lymphocytes through the purinergic pathway remains unknown. In the present study, we compared DP-MSCs, PDL-MSCs, and G-MSCs in terms of CD26, CD39, and CD73 expression; their ability to generate adenosine (ADO) from ATP and AMP; and whether the concentrations of ADO that they generate induce an immunomodulatory effect on T lymphocytes. BM-MSCs were included as the gold standard. Our results show that DT-MSCs present similar characteristics among the different sources analyzed in terms of the properties evaluated; however, interestingly, they express more CD39 than BM-MSCs; therefore, they generate more ADO from ATP. In contrast to those produced by BM-MSCs, the concentrations of ADO produced by DT-MSCs from ATP inhibited the proliferation of CD3^+^ T cells and promoted the generation of CD4^+^CD25^+^FoxP3^+^CD39^+^CD73^+^ Tregs and Th17^+^CD39^+^ lymphocytes. Our data suggest that DT-MSCs utilize the adenosinergic pathway as an immunomodulatory mechanism and that this mechanism is more efficient than that of BM-MSCs.

## 1. Introduction

Bone marrow mesenchymal stem/stromal cells (MSCs) are characterized by their regenerative and immunomodulatory properties and are exploited in clinical treatments for various diseases [1,2]. However, obtaining them from this source is a painful and invasive process for the donor, in addition to other factors such as the age and sex of the donor, which can compromise their biological potential [3,4,5,6,7]. Today, MSCs can be obtained from other sources, such as the umbilical cord blood (UCB) [8], the placenta [9], the adipose tissue, the skin [10], and dental tissues (DT-MSCs) [11]. Most of these sources of MSCs have properties similar to those of BM-MSCs and are obtained by means of less invasive processes for the donor, and some sources can even be considered biological waste. DT-MSCs, such as dental pulp (DP-MSCs), periodontal ligament (PDL-MSCs), and gingival tissue (G-MSCs), are easy to obtain, provide a high number of MSCs with high proliferation rates, and have been shown to have differentiation capacity in vitro, being able to differentiate into chondrocytes, osteoblasts, and, to a lesser extent than other sources, such as BM-MSCs, adipocytes. Additionally, they have a high capacity for neuronal differentiation and differentiation into other dental cells, and in several clinical studies, they have shown a high capacity to regenerate bone tissue and dental pulp and repair other dental tissues. Finally, they have also been suggested for regulating the immune system [12]. These cells can be considered alternative sources to BM in cell therapy; however, the mechanisms by which these MSCs modulate the immune system are not yet fully understood, and therefore it is important to elucidate their immunoregulatory mechanisms before they are considered in the clinic for therapeutic treatments [13,14].

BM-MSCs have been shown to carry out their immunosuppressive function through the secretion of soluble factors, cell-cell contact, extracellular vesicles, and the production of adenosine (ADO) from molecules such as ATP [15,16,17]. ATP is a nucleotide that is found at high levels (3–10 mM) in the extracellular environment [18,19] under inflammatory conditions. However, ATP is important for the activation and provision of energy to immune cells during inflammation, and ATP concentrations must be controlled through phosphohydrolysis, which is carried out through the action of the ectoenzyme CD39, which converts ATP into AMP, which is hydrolyzed by CD73 to generate ADO, a nucleoside with immunomodulatory properties [20].

ADO initiates its biological effects through interactions with its four receptors, A_1_, A_2_A, A_2_B, and A_3_ [20]. The binding of ADO to the A_1_ and A_3_ receptors triggers inflammatory responses, while the A_2_ receptors, which are prevalent in activated immune cells, stimulate an anti-inflammatory response by regulating cytokine production, degranulation, chemotaxis, and cytotoxicity. ADO also regulates apoptosis, cell proliferation, and promotes the differentiation of anti-inflammatory cells, such as M2 macrophages and regulatory T lymphocytes (Tregs) [20,21,22,23,24,25,26,27,28,29]. Interestingly, ADO increases the expression of CD39 and CD73 in Treg lymphocytes, which improves their anti-inflammatory potential, since they can produce greater levels of ADO as one of their immunomodulatory mechanisms compared with other populations of Tregs that do not express one or both ectoenzymes [24]. Similar effects have been observed in Th17 lymphocytes, which, under normal conditions, are inflammatory cells, but under the effect of ADO, they acquire an anti-inflammatory profile and express CD39, which is also capable of generating ADO [30]. Importantly, the levels of ADO are regulated by its conversion into inosine (INO) through a membrane ectoenzyme linked with CD26, adenosine deaminase (ADA), to reduce its immunomodulatory properties and long-term effects, which could be counterproductive for the body [31,32].

Previous data from our working group showed that DP-MSCs, PDL-MSCs, and G-MSCs have similar immunosuppressive effects, such as the ability to decrease the proliferation of CD3^+^ T lymphocytes and the production of TNF-α. These cells also increase the production of anti-inflammatory molecules, such as IL-10 and PGE_2_, and induce the generation of CD4^+^CD25^+^Foxp3^+^ Tregs in a manner similar to that of BM-MSCs [33]. However, to date, it is unknown whether DT-MSCs exploit the adenosinergic pathway as an immunomodulatory mechanism. To contribute to the knowledge of the mechanisms associated with the immunomodulatory properties of DP-MSCs, PDL-MSCs, and G-MSCs, we carried out a comparative study of these sources to determine the expression of CD26, CD39, and CD73 and their ability to generate ADO and to determine if the concentrations they produce are capable of modulating the CD3 T lymphocyte response in vitro. This is one of the first studies in which the presence of the adenosinergic pathway was compared among the three sources, which may help us to propose which source is most appropriate for immunomodulatory applications in vivo.

## 2. Results

### 2.1. DP-MSCs, PDL-MSCs, and G-MSCs Express CD26, CD39, and CD73

DT-MSC samples (*n* = 3 samples/source) were compared with BM-MSCs (*n* = 3) and were shown to preserve the characteristics established by the International Society for Cell Therapy (ISCT) for the identification of MSCs [34]. All samples expressed the markers CD105, CD90, CD73, CD13, and HLA-ABC in more than 90% of the cell population, and less than 2% were positive for the markers CD34, CD45, CD31, CD14, and HLA-DR (Appendix A). In addition, these cells exhibited fibroblast morphology (Appendix A) and multipotential differentiation toward adipocytes (lower in DT-MSCs than in BM-MSCs), chondrocytes, and osteoblasts when they were cultured in a specific inducer medium (Appendix A).

CD39 and CD73 participate in the production of ADO, while ADA, which is associated with the cell membrane through CD26, regulates the concentration of this nucleoside by converting it into INO; together, these three ectonucleases are primarily involved in the adenosinergic pathway [35]. The expression of CD39 and its ability to hydrolyze ATP to AMP and subsequently generate ADO via CD73 has been described in BM-MSCs in previous studies, as has the expression of ADA/CD26 and its ability to generate INO, suggesting that the adenosinergic pathway is one of the immunomodulatory mechanisms of these cells [15,16,17,36,37,38]. The expression of CD39 has also been confirmed in dental tissues, such as DP-MSCs and G-MSCs [39,40,41,42,43]. However, there is no comparative study of this pathway in DT-MSCs and BM-MSCs.

To determine whether DT-MSCs use the adenosinergic pathway as one of their immunomodulatory mechanisms, the percentages of CD26-, CD39-, and CD73-expressing cells were first evaluated by means of flow cytometry (Figure 1). Our results show that CD26 was expressed in 54.41 ± 11.91% of the population of BM-MSCs, while in the DP-MSCs, PDL-MSCs, and G-MSCs, it was expressed in 40.22 ± 22.3%, 38.13 ± 24.5%, and 37.51 ± 23.96% of the population, respectively; this difference was not significant (Figure 1a,c). CD39 was expressed in 43.3 ± 5.58% of the DP-MSCs, 44.81 ± 4.81% of the PDL-MSCs, and 39.71 ± 3.89% of the G-MSCs; these percentages were significantly greater (*p* < 0.05) than those in the BM-MSCs, where it was observed in 31.98 ± 3.67% of the population (Figure 1a,c). CD73 was uniformly expressed in more than 99% of the MSCs from all sources (Figure 1a,c).

We also evaluated the mean fluorescence intensity (MFI) of the three ectonucleases, and the results were normalized to those of BM-MSCs, which were assigned a value of 1, as a reference. The MFIs of CD26 in DP-MSCs, PDL-MSCs, and G-MSCs were 0.20 ± 0.1, 0.21 ± 0.09, and 0.24 ± 0.08, respectively, which were significantly lower than those in BM-MSCs (*p* < 0.05). In contrast, the expression of CD39 (DP-MSCs = 5.44 ± 3.26; PDL-MSCs = 7.51 ± 3.99; G-MSCs = 5.32 ± 2.74) and CD73 (DP-MSCs = 1.35 ± 0.38; PDL-MSCs = 1.35 ± 0.33; G-MSCs = 1.44 ± 0.25) in DT-MSCs was greater than that in BM-MSCs; however, the difference was not significant (Figure 1b,c). These results indicate that DT-MSCs express three ectonucleases involved in the adenosinergic pathway and they may therefore generate ADO from ATP and AMP in the same way as those of BM-MSCs.

### 2.2. DT-MSCs Produce ADO from ATP or AMP via CD39 and CD73

The adenosinergic pathway contributes significantly to the immunosuppressive capacity of BM-MSCs, particularly through the participation of the CD39 and CD73 ectonucleotidases, which generate concentrations of ADO from ATP capable of modulating the response of immune cells such as T lymphocytes [15,16]. After the expression levels of CD39 and CD73 in DT-MSCs and BM-MSCs were analyzed, their abilities to phosphohydrolyze ATP and AMP in ADO were compared. For this, 2 × 10^6^ MSCs from each source were cultured with 5 mM ATP or AMP for 5 h, and the conditioned media (CM) was analyzed via TLC. The production of ADO from both nucleotides was observed in the CM of all of the MSC sources (MSC-CM) (Figure 2, circles). To determine whether CD39 and CD73 are responsible for the production of ADO, specific inhibitors of CD39 (POM-1) and CD73 (APCP) were used, and we observed that, under both conditions, there was no production of ADO (Figure 2).

ADA is the ectoenzyme that converts ADO to INO, and its expression is correlated with the presence of CD26 since this protein is anchored to the cell membrane [31,32]. Given that all sources of MSCs express CD26, we determined whether INO was present in the CM of MSCs. To do this, MSCs were grown with ATP or AMP and maintained for 24 h. The CM were collected for evaluation, and the presence of INO was observed in the CM of all of the MSC cultures through TLC (Figure 3, circles). Interestingly, we observed through the intensity of the bands of this nucleoside that the concentration of INO was lower than that of ADO. These results suggest that the low expression of ADA in all sources of MSCs transforms ADO into INO.

Through TLC, it was possible to qualitatively observe that the MSC-CM obtained after 5 h of culture had a similar concentration of AMP-derived ADO from all sources, while the generation of ATP-derived ADO was similar in all of the DT-MSC-CM and lower in BM-MSC-CM (Figure 2). To confirm this observation, the ADO concentrations in the MSC-CM were evaluated through UPLC (Figure 4). All of the MSC-CM cultured with AMP generated similar concentrations of ADO (BM-MSC-CM = 482.84 ± 121.63; DP-MSC-CM = 517.2 ± 54.75; PDL-MSC-CM = 505.95 ± 38.72; G-MSC-CM = 484.09 ± 77.25 μM/mL). On the other hand, in DT-MSC-CM, where ADO was generated from ATP, similar concentrations of ADO were detected, and this was corroborated in BM-MSCs cultured with ATP, in which the ADO concentration was significantly lower (BM-MSC-CM = 72.13 ± 101.15; DP-MSC-CM = 444.17 ± 68.23; PDL-MSC-CM = 400.78 ± 79.84; G-MSC-CM = 371.02 ± 89.52 μM/mL; *p* < 0.05). In the CM of the cultures in the absence of substrate or in the presence of the ectonucleotidase inhibitors, no ADO was detected (Figure 4). These results indicate that DT-MSCs are more efficient at producing ADO from ATP than BM-MSCs are.

### 2.3. The Concentration of ADO Generated by DT-MSCs Decreases the Proliferation of CD3^+^ Lymphocytes

ADO is a molecule with anti-inflammatory properties that mediates its biological effect through interactions with the A_2_A and A_2_B receptors, and its activation affects various functions of immune cells, such as the proliferation and secretion of proinflammatory cytokines [20]. Our group showed that the concentrations of ADO present in MSC-CM from cervical cancer patients inhibit the proliferation of T lymphocytes through the A_2_A receptor [44]. For this reason, we evaluated whether the concentration of ADO present in DT-MSC-CM is capable of decreasing the proliferation of CD3^+^ T lymphocytes. For this purpose, MSC-CM from different sources where ADO was generated after 5 h were diluted to a final concentration of 20% and added to activated CD3^+^ T lymphocyte cultures. Compared to the proliferation control group (see Materials and Methods, Section 4.8), the proliferation of lymphocytes in the different DT-MSC-CM groups cultured with ATP or AMP decreased by 25% (*p* < 0.05) (Figure 5a,b). Similar values were observed in the cultures with BM-MSC-AMP-CM, as well as those obtained with pure ADO [1000 μM]. In the cultures with BM-MSC-ATP-CM, a nonsignificant decrease of 10% in the proliferation of lymphocytes was observed compared with that in the control group (Figure 5a,b). By inhibiting A_2_A and A_2_B receptors on CD3^+^ T lymphocytes and subsequently adding MSC-ATP-CM or MSC-AMP-CM, the percentage of cellular proliferation was similar to that of the control. Similarly, in the cultures with MSC-ATP+POM-1-CM or with MSC-AMP+APCP-CM, no decrease in the percentage of CD3^+^ lymphocyte proliferation was observed (Figure 5a,b). These results indicate that, unlike BM-MSCs, DT-MSCs in the presence of ATP generate concentrations of ADO that significantly decrease the proliferation of T lymphocytes.

### 2.4. ADO Produced by DT-MSCs Induces the Generation of CD4^+^CD25^+^FoxP3^+^ Tregs and Induces the Coexpression of CD39^+^CD73^+^

The ADO could be involved in the generation of Treg lymphocytes. For this reason, we evaluated whether the amount of ADO produced by DT-MSCs induced the generation of CD39^+^CD73^+^ Tregs (Figure 6a–c). To this end, purified CD4^+^ T lymphocytes were cultured and activated, MSC-CM were added, and the immunophenotype was determined by means of flow cytometry. Treg-inducing medium (IL-2 + TGF-β1) and pure ADO [1000 μM] were added as control conditions. With the exception of BM-MSC-ATP-CM, DT-MSC-CM increased the percentage of CD4^+^CD25^+^FoxP3^+^ Tregs (Figure 6b) and the percentage of Tregs that coexpressed CD39^+^CD73^+^ (Figure 6a,b), in a similar way to the controls. Interestingly, we observed that this increase in the percentage of Tregs and the expression of CD39 and CD73 decreased when A_2_ receptors were blocked in lymphocytes, and values similar in those cultures in the presence of ATP+POM-1-CM and AMP+APCP-CM were observed (Figure 6b,c, respectively). These results suggest that, unlike that produced by BM-MSCs, the concentration of ADO produced by DT-MSCs in the presence of ATP is involved in the ability of DT-MSCs to induce the differentiation of Treg lymphocytes and increase the coexpression of CD39 and CD73 in these cells. However, the ability of DT-MSCs and BM-MSCs to induce Tregs is similar when they produce ADO from AMP.

### 2.5. The ADO Produced by DT-MSCs Induces the Expression of CD39 in Th17 Lymphocytes

MSCs are known to inhibit Th17 lymphocyte differentiation through various mechanisms, such as anti-inflammatory molecules, extracellular vesicles, and mitochondrial transfer [45]. Some studies have even suggested that BM-MSCs and G-MSCs could decrease this population through ADO [16,41,42,46]. However, this effect is still controversial given that there is evidence that the ability to produce ADO does not decrease the differentiation of Th17 lymphocytes but does induce anti-inflammatory characteristics, including the expression of CD39 [30,47]. Therefore, we evaluated the effect of DT-MSC-induced ADO on the population of Th17 lymphocytes. For this purpose, purified CD4^+^ T lymphocytes were cultured, activated, and differentiated with the cytokines IL-1β + IL-6 + IL-23 + TGF-β1. MSC-CM were added, and the immunophenotype and expression of CD39 were evaluated through flow cytometry (Figure 7a–c).

We observed a significant decrease in the percentage of Th17^+^ cells in the cultures in the presence of MSC-ATP-CM or MSC-AMP-CM in which ADO was generated compared with that in the control of Th17 cells, except in the cultures where BM-ATP-CM were added (Figure 7b). Interestingly, under these same conditions, except for those with BM-MSC-ATP-CM, we observed a significant increase (*p* <0.05) in the percentage of the Th17^+^CD39^+^ population with respect to that of the control population, whose value was similar to that generated in the presence of pure ADO (Figure 7a,b). In lymphocytes cultured in the presence of A_2_ receptor inhibitors, the percentage of the Th17 population did not decrease (Figure 7a,b), nor did the percentage of the Th17^+^CD39^+^ population increase (Figure 7a,c). These results suggest that the amount of ADO present in DT-MSC-CM decreases the differentiation of Th17 lymphocytes and induces the expression of CD39 in these cells.

## 3. Discussion

In recent years, the biological properties of MSCs have attracted the attention of researchers worldwide due to their potential application in the treatment of different diseases, such as immunological diseases. At present, the BM represents the main source of MSCs; however, due to certain inconveniences in obtaining them, more accessible alternative sources have been sought [4]. In this regard, we recently published a review describing the biological characteristics that position dental tissues as excellent sources for obtaining MSCs and that, due to their properties, make them very promising candidates for cell therapy [12].

At the clinical level, one of the aspects to consider is that, in order to use MSCs in cell therapy treatments and depending on the disease, it has been calculated that between 5 and 10 million MSCs per kilogram of patient weight are needed, but it is also necessary to consider the route and frequency of administration. Therefore, ex vivo expansion procedures are of vital importance to obtain sufficient numbers of MSCs, and experimental procedures for this purpose are still under development. Indeed, one of the key aspects of such procedures is to prevent MSCs from losing their properties after in vitro expansion. Currently, ex vivo expansion protocols have advanced considerably and the negative effects of large-scale expansion have been reduced; protocols that not only prevent MSCs from losing their immunomodulatory effects but can also enhance them have even been developed [48,49,50,51].

In the case of DT-MSCs, few studies have evaluated the effect of large-scale expansion; however, small-scale cultures have been developed using Xeno-free culture medium and microcarriers that increase the surface area for cell expansion, conditions that are used in large-scale cultures. Interestingly, these studies have found that some characteristics, such as the morphology, immunophenotype, and differentiation capacity, are not altered after expansion [52,53,54,55]; however, it will be important to further investigate the effects that these protocols may have on the immunomodulatory properties of MSCs in the future. Despite our knowledge of the biological properties of DT-MSCs, it is necessary to elucidate the molecular mechanisms by which they regulate immunity in different settings, so it is essential to carry out more detailed studies before considering their use in clinical protocols.

In this study, samples obtained and previously cryopreserved by our working group were used, and the properties established by the ISCT for the characterization of MSCs were evaluated [34]. In this regard, we observed that the adherent cell populations obtained displayed the biological characteristics of MSCs, including fibroblast morphology, immunophenotype markers, osteogenic and chondrogenic capacity, and low adipogenic capacity. These results agree with what was previously reported by our group [33].

BM-MSCs have the ability to modulate the immune system through the secretion and expression of different immunosuppressive molecules, the secretion of extracellular vesicles, and the generation of metabolites; these characteristics have been reported by different working groups, including our group [56,57]. We recently demonstrated that DP-MSCs, PDL-MSCs, and G-MSCs reduce the proliferation of T lymphocytes through the secretion and expression of anti-inflammatory molecules and induce the differentiation of Tregs in a manner similar to that of BM [33]. In the last decade, it has been observed that BM-MSCs use the adenosinergic pathway as one of their immunomodulatory mechanisms [15,16,58]. Through this pathway, ATP is phosphorylated in ADOs through the co-expression of the ectonucleotidases CD39 and CD73. Although CD73 is a characteristic marker of MSCs, few studies have demonstrated the expression of CD39 in these cells. In 2011, Sattler et al. demonstrated the presence of CD39 in mouse BM-MSCs [58]. Its expression was subsequently confirmed in human BM-MSCs [15], and in fact, our group showed that CD39 is expressed in MSCs from the normal cervix and cervical cancer [44], while others have shown its presence in limbal and dermal tissue [59] and pharynx and larynx and head and neck cancer [60].

When evaluating the expression of CD39 in DP-MSCs, PDL-MSCs, and G-MSCs, we found a greater percentage of CD39^+^ cells compared to BM-MSCs, and in the same way, according to the MFI values, we observed a greater percentage of CD39 molecules in DT-MSCs than in BM-MSCs. However, there was high variation between the samples from each source, which indicates that the expression of this ectonuclease is heterogeneous and could be regulated by the differential expression of proinflammatory cytokines, hypoxia, and oxidative stress, among other factors [61]. In BM-MSCs, no expression of this ectoenzyme has been reported [62]; however, it has also been detected in 15–84% of the cell population [15,16,17], while in dental tissues, such as DP-MSCs, CD39 occurs in between 1% and 10% of the cell population [39,40], and in G-MSCs, it occurs in between 27% and 50% of the cell population [41,42,43]. In fibroblasts derived from the periodontal ligament, the expression of this ectoenzyme has also been reported [63]. In addition, in vitro tests revealed that CD39 expression is increased when MSCs are cultured in medium supplemented with different nutrients or with different concentrations of FBS, and even when they are cultured with other cells, such as T lymphocytes [15,39,63]. Similarly, we previously demonstrated that MSCs from tumor tissues have greater expression of CD39 than their counterparts derived from healthy tissue [44]. These data suggest that various factors increase the expression of CD39 in MSCs.

After the expression of CD39 and CD73 in the DT-MSCs was determined, their ability to hydrolyze ATP or AMP to ADO was evaluated, and we found that all the sources produced ADO from both nucleotides. Interestingly, we observed that DT-MSCs generated more ADO from ATP than BM-MSCs, which is consistent with the detection of a greater percentage of the population that expresses CD39 in DT-MSCs, even with the increase in CD39 molecules at the cellular level with respect to BM-MSCs. These results are consistent with previous studies provided by our group and other groups, where the hydrolysis capacity of MSCs derived from tumors and MSCs from healthy tissues was compared, and the former generated more ADO from ATP since they express higher levels of CD39 [44,60]. On the other hand, the amount of ADO generated from AMP was similar among all the sources since the expression of CD73 was uniform. When CD39 or CD73 was blocked with specific inhibitors, no ADO was detected, which indicates that the production of this nucleoside was due to the enzymatic activity of these ectonucleases.

The half-life of ADO is approximately 10 s [64,65], and its levels are reduced due to the activity of ADA, an enzyme that is expressed intracellularly or on the cell surface upon binding to CD26. ADA catalyzes the irreversible deamination of ADO, allowing the formation of INO and thus reducing its anti-inflammatory effect [31]. Interestingly, we found that MSCs from all sources generated INO, but this effect could only be observed after 24 h or longer in culture; therefore, we evaluated the expression of CD26 in the cell membrane since its expression is associated with the presence of ADA [32]. We found that DT-MSCs express CD26 in 50% of the cell population; however, the ability of DT-MSCs to generate INO was also reduced. Few studies have evaluated the presence of CD26 or the direct activity of ADA in MSCs, such as those from head and neck cancer [60], BM, UCB, and adipose tissue [36,38]. In these studies, it was observed that the amount of INO that these MSCs generate depends on the expression levels for this enzyme, which is consistent with our findings. On the other hand, other possible factors involved in why we find low levels of INO in CM must be estimated, it is known that cells are capable of transporting ADO and INO from the inside to the outside and vice versa through nucleoside transporters, this based on intra- and extracellular concentrations [66,67], therefore, we can consider the possibility that the ADO produced by MSCs is transported to the interior of these cells and converted into INO by intracellular ADA and subsequently released into the extracellular medium, therefore, the presence of intracellular ADA in DT-MSCs and its catalytic activity must be determined in future studies to determine if it plays an important role in the regulation of ADO concentrations.

ADO can decrease the proliferation of immune cells through the A_2_A and A_2_B receptors [20,24]. Therefore, we evaluated whether the concentration of ADO present in DT-MSC-CM decreased the proliferation of T cells. DT-MSC-CM was added to activated lymphocytes, and the proliferation decreased by 25%, similar to that of pure ADO [1000 μM]. The decrease in proliferation was lower in the BM-MSC-ATP-CM group than in the DT-MSC-CM groups, which indicates a lower capacity of the BM-MSCs to generate ADO. We also determined that the immunosuppressive effect of ADO occurs through A_2_ receptors, since when T lymphocytes are blocked with specific inhibitors, we do not observe a decrease in proliferation. Finally, considering that MSCs secrete a wide variety of anti-inflammatory molecules, such as IL-10 [68,69,70,71], TGF-β [72,73,74], prostaglandin E_2_ [75,76,77], nitric oxide [78,79], and indolamine 2,3-dioxygenase [80,81,82], and even secrete extracellular vesicles with anti-inflammatory capacity [83], we evaluated DT-MSC-CM without any substrate or where ATP or AMP had been added in the presence of CD39 or CD73 inhibitors, and we observed that these CM did not affect the proliferation of T lymphocytes. These experimental findings confirm that the ADO produced by DT-MSCs was responsible for decreasing the proliferation of T lymphocytes.

ADO can promote the differentiation of lymphocyte Tregs [84], and this differentiation has also been observed through the use of specific agonists of A_2_ receptors [25,26,27,84]. Interestingly, these cells express relatively high levels of inhibitory molecules such as LAG-3 and CTLA-4 and relatively high levels of CD39 and CD73 [24]. Tregs that coexpress both ectonucleases are considered to have greater anti-inflammatory effects than those that do not, since they generate ADO as one of their immunomodulatory mechanisms [85,86]. In addition, CD39 increases the stability of Tregs in the presence of inflammatory cytokines, preventing them from being redifferentiated to a proinflammatory phenotype [87]. For this reason, we evaluated whether the ADO present in DT-MSC-CM could induce the differentiation of Tregs from CD4^+^ cells and whether these cells would have increased expression of CD39 and CD73. Our results show that the ADO present in DT-MSC-ATP-CM and DT-MSC-AMP-CM increased the population of CD4^+^CD25^+^FoxP3^+^ Tregs similarly to the control of Tregs generated from the classical cytokines IL-2 + TGF-β1 and the control of pure ADO [1000 μM] and, furthermore, that the percentage of Tregs that coexpressed CD39^+^CD73^+^ also increased in a similar way to the controls. Interestingly, no similar results were observed for BM-MSC-ATP-CM. Therefore, we demonstrated that DT-MSCs have greater potential to generate Tregs with the CD4^+^CD25^+^FoxP3^+^CD39^+^CD73^+^ immunophenotype and that this effect is inhibited by blocking A_2_ receptors on CD4 lymphocytes, which indicates that this effect is mediated through ADO.

Another immunomodulatory effect of ADO is its ability to inhibit the differentiation of some subpopulations of T lymphocytes and the secretion of their characteristic cytokines, as in the case of Th1 and Th2 lymphocytes [88]. However, the effect of ADO on other cell subpopulations, such as Th17 lymphocytes, has rarely been studied. In some animal models, ADO decreases the differentiation of this population and the secretion of IL-17 [89,90]. However, other studies have shown otherwise [47,91,92,93]. In a recent article, it was observed that ADO stimulates the differentiation of Th17 lymphocytes and the secretion of IL-17 in human lymphocytes. However, these Th17 cells express anti-inflammatory genes and produce IL-10; they also express CD39, and when cultured with activated T lymphocytes, they are capable of reducing their proliferation and activation, and they are called Th17 suppressors [30]. For this reason, we evaluated the effect of DT-MSC-induced ADO on Th17 lymphocytes. Our results showed that CM decreased the differentiation of Th17 lymphocytes in a similar way to that of the pure ADO control [1000 μM]. Interestingly, a greater percentage of Th17 cells that were generated in the presence of ADO were positive for CD39 compared to those not stimulated with this nucleoside. This effect was observed for all the MSC-CM except for BM-MSC-ATP-CM. This ADO-induced increase in CD39 is consistent with previous reports [30]. Previous studies have shown that Th17^+^CD39^+^ lymphocytes are capable of generating ADO and have high anti-inflammatory potential [94]. Therefore, the generation of these cells and CD39^+^CD73^+^ Tregs by the ADO produced by DT-MSCs could improve the anti-inflammatory microenvironment induced by MSCs.

## 4. Materials and Methods

### 4.1. Isolation and Culture of Mesenchymal Stromal Cells (MSCs)

The samples of BM-MSCs, DP-MSCs, PDL-MSCs, and G-MSCs (*n* = 3/source) in R4-R6 were obtained from a cryopreservation bank in the mesenchymal stem cell laboratory of the IMSS, Siglo XXI.

#### 4.1.1. Isolation of BM-MSCs

BM-MSCs were obtained with the informed consent of hematologically healthy adult volunteer donors according to the guidelines of the IMSS Hospital of Traumatology and Orthopedics as previously described [95]. Briefly, samples were obtained from BM aspirates from the iliac crest and placed in a 50 mL tube (Corning, New York, NY, USA) with 15 mL of HyClone Roswell Park Memorial Institute (RPMI)-1640 Me-dium (Cytiva, Marlborough, MA, USA) containing 10% fetal bovine serum (FBS, Gibco BRL, Rockville, MD, USA) and then washed with phosphate-buffered solution (PBS, Thermo Fisher Scientific, Waltham, MA, USA). A density gradient was prepared with Ficoll-Paque Plus, density 1.077 + 0.001 g/mL (GE Healthcare Bio-Sciences AB, Uppsala, Sweden). The cells were centrifuged at 300g for 30 min, and the interface was washed with PBS containing 3% FBS and 1 mM EDTA. The mononuclear cell (MNC) pellet was resuspended in HyClone Dulbecco’s Modified Eagle’s Medium with low glucose (lg-DMEM; Cytiva) supplemented with 15% FBS. The total number of nucleated cells and their viability were determined by counting with Tuerk´s solution and trypan blue (Sigma-Aldrich, St. Louis, MO, USA), respectively. 5–10 × 10^6^ MNCs were seeded in a 100 mm Petri dish (Corning) and incubated at 37 °C with 5% CO_2_. After four days, a PBS wash was performed to remove non-adherent cells, changing the medium twice per week. Upon reaching a confluence of 80% or greater, the cells were trypsinized (0.05% trypsin, 0.53 mM EDTA; Gibco) for 5 minutes at a temperature of 37 °C with 5% CO_2_, then, total number of cells and the viability were determined by trypan blue. Finally, 1 × 10^6^ MSCs were cryopreserved in 2 mL cryovials (Corning) embedded in freezing medium added with 50% FBS and 10% dimethyl sulfoxide (Sigma-Aldrich) and protected in liquid nitrogen until use.

#### 4.1.2. Isolation of DT-MSCs

On the other hand, DT-MSCs were obtained from healthy donors in the age range of 25 to 28 years, who underwent prophylactic third molar extraction at the Maxillofacial Surgery Clinic of the Division of Postgraduate Studies and Research at the School of Dentistry of the Universidad Nacional Autónoma de México. Patients were informed about the study prior to obtaining their consent and donation of extracted dental organs, and the protocol was approved by the Research and Ethics Committee of the School of Dentistry (CIE/1110/2017), DT-MSCs were obtained as previously described [33]. Briefly, after third molar extraction surgery, the tooth was transferred to a 50 mL tube with 5 mL of Hyclone lg-DMEM supplemented with 15% FBS and 100 IU/mL penicillin. The samples were processed immediately after extraction in a 100 mm Petri dish. The periodontal ligament covering the roots of the tooth and the gingival tissue which was firmly attached to the periosteum were dissected; finally, the tooth was transversely sectioned with a diamond disc to expose the pulp cavity and thus extract the dental pulp. The three tissues were mechanically disaggregated in isolation and placed in a 6-well plate (Corning), embedded in 1 mL of HyClone Alpha-Modified Minimum Essential Medium (α-MEM; Cytiva) supplemented with 10% FBS, 2 mM L-glutamine, 100 IU/mL penicillin, 100 μg/mL streptomycin, 100 μg/mL gentamicin (all from Gibco) and maintained for 2 to 5 weeks, changing the medium twice per week. Upon reaching a confluence of 80% or greater, the cells were trypsinized and cryopreserved until use.

All MSCs sources were maintained under standard conditions for expansion and subsequent use in each experiment involving them. Briefly, they were cultured in HyClone lg-DMEM supplemented with 10% FBS, 2 mM L-glutamine, 100 U/mL penicillin, 100 mg/mL streptomycin, and 100 mg/mL gentamicin (all from Biowest, Riverside, MO, USA) at 37 °C with 5% CO_2_ and saturating humidity. Upon reaching a confluence of 80% or greater, the cells were trypsinized and number of cells and viability were determined by trypan blue staining.

### 4.2. Cytometry and Immunophenotyping of DT-MSCs and BM-MSCs

For the experiments where flow cytometry was used, the supplier’s instructions for staining with each antibody were followed. For extracellular labeling, cells were resuspended in 100 μL of PBS supplemented with 1 mM EDTA, and 25 μL of the viability marker Ghost Dye Red 780 (TONBO Biosciences, San Diego, CA, USA; 1 μL dissolved in 1499 μL of PBS) was added for 15 min at room temperature. Subsequently, the membrane was blocked with FBS for 15 min at 4 °C, and the appropriate antibodies were added for 20–30 min. The cells were then fixed with FACS Lysing Solution (Becton Dickinson, Biosciences; San José, CA, USA). For the marking of intracellular molecules, the cultures were supplemented with Golgi Stop (Becton Dickinson) for 5 h, after which the cells were permeabilized with a FoxP3 staining buffer kit (Invitrogen, Waltham, MA, USA) following the manufacturer’s instructions. The samples were analyzed on a Cytek Aurora three-laser spectral cytometer (Cytek Biosciences, Fremont, CA, USA), and at least 10,000 events were acquired. The data were analyzed with FlowJo v10 software (Becton Dickinson).

To determine the immunophenotype of the samples, at least 1 × 10^5^ cells were in-cubated with the following fluorochrome-linked monoclonal antibodies: anti-human CD105-BV421; anti-human CD90-PE-Cy5; anti-human CD73-PE-Cy7; anti-human CD13-APC; anti-human CD34-APC; anti-human HLA-ABC-FITC; anti-human CD31-FITC; anti-human CD45-PE; anti-human CD14-PE (all from Becton Dickinson); and anti-human HLA-DR-PE-Cy7 (BioLegend, San Diego, CA, USA). The samples were evaluated by means of flow cytometry.

### 4.3. Morphology and Induction of Differentiation of DT-MSCs and BM-MSCs

To evaluate the morphology, the cells were stained with Wright’s stain (Sigma-Aldrich) and observed using a phase contrast microscope (Zeiss, Oberkochen, Germany).

To induce the differentiation of cells into adipocytes, chondrocytes, and osteoblasts, the protocols described above were followed [95]. Briefly, the cells were cultured in a specific inducer medium for each lineage (Stem Cell Technologies, Vancouver, CO, Canada) for 3 weeks, and the medium was changed twice a week. The chondrogenic inducer medium was supplemented with 10 ng/mL TGF-β3 (PeproTech, Cranbury, NJ, USA). Adipogenic differentiation was evaluated with Oil Red O staining (Sigma-Aldrich) to detect intracellular lipid vacuoles.

Chondrogenic differentiation was determined with Alcian blue staining (Sigma-Aldrich) for the detection of mucopolysaccharides.

Osteogenic differentiation was assessed with a SIGMA Fast BCIP/NBT (5-bromo-4-chloro-3-indolyl-phosphate/nitro blue tetrazolium; Sigma-Aldrich) stain to determine alkaline phosphatase activity.

### 4.4. Expression of CD26, CD39, and CD73 in DT-MSCs and BM-MSCs

To determine the expression of the ectonucleotidases CD26, CD39, and CD73 in the cell membrane, the MSCs were labeled with the following fluorochrome-linked monoclonal antibodies: anti-human CD26-PE (BioLegend); anti-human CD39-APC; and anti-human CD73-PE-Cy7 (both from Becton Dickinson), and subsequently, the cells were analyzed by means of flow cytometry.

### 4.5. Phosphohydrolytic Activity of CD39/CD73 by Silica Gel Thin Layer Chromatography (TLC)

To determine the enzymatic activity of CD39 and CD73 in DT-MSCs, the method described above by our group was used with some modifications [44]. Briefly, 2 × 10^6^ viable MSCs were cultured in a 1000 μL volume of StemLine II medium (Sigma-Aldrich) supplemented with or without 5 mM ATP or AMP (both from Sigma-Aldrich). In some cultures, 10 μM Sodium polytungstate (POM-1) or 10 μM α,β-Methyleneadenosine 5′-diphosphate (APCP) (both from Sigma-Aldrich) was used to inhibit the ectonucleotidases CD39 and CD73. Then, the CM were collected at 5 hours and frozen at −70 °C until use.

To perform TLC, a 1 μL aliquot of each CM sample was taken and placed on a TLC plate (Sigma-Aldrich). To determine the degradation products of each of the substrates, pure standards of ATP, ADP, AMP, INO, and ADO (all from Sigma-Aldrich) were prepared at a concentration of 5 mM using Milli-Q water as the solvent. The samples placed on the TLC plate were separated into a mobile phase composed of isobutanol/isoamyl alcohol/ethoxyethanol/ammonia/water (9:6:18:9:15). The hydrolysis of the ADO nucleotides was visualized and photographed on a transilluminator with UV light.

### 4.6. Ultraperformance Liquid Chromatography (UPLC)

An UPLC system (UPLC Acquity, Waters Corporation, Milford, MA, USA) was used to quantify the amount of ADO present in the MSC-CM. Quantitative analysis of samples using standard quantities of synthetic ADO was carried out with Empower 3 software (Waters, USA). The mobile phase consisted of 0.5% acetonitrile, 5% methanol, and 94.5% sodium acetate buffer, 0.25 M, pH 6.3. Volumes of 100 μL of supernatants obtained after 5 h of culture with MSCs were diluted in 400 μL of mobile phase, then centrifuged at 13,000 rpm in microtubes with Amicon membranes with a cut-off of 3000 Da to filter them, and subsequently diluted to 1:200 with the mobile phase mixture. The run conditions were as follows: a flow rate of 1 mL/min, UV detection at 254–260 nm, 2 min of retention time, room temperature, and a LiChrosfer 5 μm RP-18e 100 A (size 125 mm × 4 mm, 5 μm particle size) reverse phase column. ADO was quantified by comparing the retention time of the sample with that of the synthetic ADO used as a standard.

### 4.7. Obtaining Mononuclear Cells from Peripheral Blood and Enrichment of CD3^+^ and CD4^+^ T Lymphocytes

MNCs from peripheral blood from healthy donors were obtained by density gra-dient using Lymphoprep (δ = 1.077 g/mL, STEMCELL Technologies, Vancouver, CO, USA) and were main-tained in Hyclone RPMI-1640 supplemented with 10% FBS under standard culture conditions until use. CD3^+^ or CD4^+^ T lymphocytes were selected with a specific anti-body directed against CD3 or CD4, respectively, conjugated to magnetic beads and MS Mac columns following the supplier’s instructions (Miltenyi Biotec, Bergisch Gladbach, Germany).

### 4.8. Effect of ADO Produced by DT-MSCs and BM-MSCs on T Lymphocytes

In the different tests carried out to determine the effect of the ADO produced by DT-MSCs on T cells, lymphocytes from 3 different healthy donors were used, and the cells from each donor were evaluated with the CM of each sample (*n* = 3 samples) from each source, resulting in a total of 9 repetitions for each source. First, 1 × 10^5^ CD3^+^ or CD4^+^ T lymphocytes/well were cultured in 24-well plates with 1000 μL of HyClone RPMI-1640 medium supplemented with 10% FBS and with 20% conditioned media (CM) from MSCs (MSC-CM), where ADO was generated from ATP (ATP-CM) or AMP (AMP-CM); MSC-CM was grown in only medium (basal medium), supplemented with ATP+POM-1 (ATP+POM-1-CM), or supplemented with AMP+APCP (AMP+APCP-CM); and finally, MSC-CM, where ADO was generated from ATP (ATP-CM+Inh A_2_) or AMP (AMP-CM+Inh A_2_) + 2 μM ZM 241385, a selective antagonist of A_2_A receptor, +2 μM MRS 1754, a selective antagonist of A_2_B receptor, (Both from Sigma-Aldrich). T lymphocytes were stimulated with CD2/CD3/CD28 activation beads (Miltenyi Biotec) at a 1:1 ratio.

#### 4.8.1. Effect of DT-MSC-Induced ADO on T Lymphocyte Proliferation

Proliferation was evaluated by adding 5 μM 5(6)-Carboxyfluorescein diacetate N-succinimidyl ester (CFSE; Sigma-Aldrich) to CD3^+^ T lymphocytes, following the supplier’s instructions. The cells were cultured for 5 days, collected, labeled with anti-human CD3-APC (Becton Dickinson) and evaluated via flow cytometry. As a reference, activated lymphocytes without any stimulus were used as control group for proliferation, as well as pure ADO [1000 μM/mL] to evaluate the antiproliferative effect.

#### 4.8.2. Generation of FoxP3^+^CD39^+^CD73^+^ Treg Lymphocytes with ADO Produced by DT-MSCs and BM-MSCs

The generation of Tregs was evaluated in CD4^+^ T lymphocyte cultures supplemented with MSC-CM. As a control for the generation of Tregs, 20 ng/mL of IL-2 and 20 ng/mL of TGF-β1 were added (both from PeproTech) [96] or pure ADO [1000 μM/mL]. The cultures were kept for 4 days. To evaluate the Treg population, the cells were labeled with the following fluorochrome-linked extracellular monoclonal antibodies: anti-human CD4-PE-Cy5; anti-human CD25-FITC; anti-human CD39-APC; anti-human CD73-PE-Cy7 (all from Becton Dickinson); and with the intracellular molecule anti-human FoxP3-PE (Invitrogen), and the cells were analyzed via flow cytometry. 

#### 4.8.3. Effect of ADO Produced by DT-MSCs and BM-MSCs on Th17 Lymphocytes

To evaluate the effect of ADO on Th17 cells, CD4^+^ T lymphocytes were cultured, and differentiation was induced using 5 ng/mL TGF-β1, 10 ng/mL IL-6, 10 ng/mL IL-23 and 5 ng/mL IL-1β (all from PeproTech) [30]. MSC-CM was added, and pure ADO [1000 μM/mL] was used as a control. The cells were cultured for 4 days. To evaluate the Th17 immunophenotype, the cells were labeled with the following fluorochrome-linked extracellular monoclonal antibodies: anti-human CD4-PE-Cy5; anti-human CD39-APC (both from Becton Dickinson) and the intracellular molecule anti-human IL-17a-PE (BioLegend) and analyzed via flow cytometry.

### 4.9. Statistical Analysis

Statistical analysis was performed using GraphPD Prism 10 software with one-way ANOVA. Differences were considered to be statistically significant at *p* < 0.05. The data are presented as the mean value ± SD.

## 5. Conclusions

In this study, we compared the presence of the adenosinergic pathway in DP-MSCs, PDL-MSCs, G-MSCs, and BM-MSCs. We observed that DT-MSCs from the three sources expressed more CD39 than BM-MSCs, indicating that DT-MSCs can generate higher concentrations of ADO from ATP than BM-MSCs can. Furthermore, we demonstrate that MSCs from the sources studied express CD26 and are capable of generating INO from ADO, which indicates that this enzymatic process participates in the regulation of ADO concentrations so as not to be harmful in the long term. It is important to explore this mechanism to determine whether reducing the concentration of ADO affects the anti-inflammatory potential of DT-MSCs.

On the other hand, the concentrations of ADO generated by DT-MSCs decreased the proliferation of CD3^+^ T lymphocytes and promoted the generation of subpopulations with anti-inflammatory potential, CD4^+^CD25^+^FoxP3^+^CD39^+^CD73^+^ Tregs and Th17^+^CD39^+^ lymphocytes. Th17 and Treg lymphocytes are known to play opposite roles in immunity since, while Th17 cells participate in inflammation, Tregs maintain immune homeostasis. In autoimmune diseases, the balance between these two cell populations is disrupted, Treg levels tend to decrease, and Th17 levels increase considerably, improving inflammation. Current treatments for diseases such as psoriasis, inflammatory bowel disease, rheumatoid arthritis, and multiple sclerosis, among others, seek to restore this balance between proinflammatory and regulatory populations to control the disease [97]. Our study is the first to show that DT-MSCs could have greater potential to restore this balance than BM-MSCs, particularly through the adenosinergic pathway, which favors the generation of regulatory populations with anti-inflammatory effects.

DT-MSCs could be proposed as alternative sources to BM-MSCs and have potential for clinical application in the treatment of inflammatory diseases. This is because their immunomodulatory properties are similar to BM-MSCs, and they also present some advantages, such as their easy obtaining and greater proliferation, among others. Although most of the studies evaluating the anti-inflammatory properties of various DT-MSCs in autoimmune diseases have been limited to preclinical trials, these have shown satisfactory results in several aspects, for example, they have considerably reduced morbidity and mortality and have promoted the regeneration of affected tissues in the various study models developed. Interestingly, these preclinical trials have also included models of periodontitis and other bone defects of the oral cavity. In these trials, it was also observed that DT-MSCs were able to reduce the ravages of the disease and induce bone regeneration; this was attributed to the ability of DT-MSCs to reduce the infiltration of inflammatory cells and induce the generation of regulatory cells such as M2 macrophages and Tregs lymphocytes [12].

Additionally, several clinical studies that use these sources of MSCs have been developed, and although most have been limited to the regeneration of dental tissues in diseases such as periodontitis, their application has been effective and they have not shown serious complications, which indicates that their use is safe, however, the mechanisms related to their effectiveness in these trials have not been discerned, so it is important to emphasize these aspects before ensuring that they are feasible sources to replace BM-MSCs [12].

Finally, determining whether DT-MSCs utilize the production of ADO as an immunoregulatory mechanism is crucial to proposing them as alternative sources in future clinical protocols, due to the enormous immunomodulatory potential of this molecule, which will possibly act effectively in different inflammatory scenarios together with the other immunomodulatory mechanisms of DT-MSCs. Improving our understanding of the adenosinergic pathway is a major challenge for the near future.

In summary, our study indicated that DP-MSCs, PDL-MSCs, and G-MSCs use the adenosinergic pathway as one of their immunomodulatory mechanisms in a similar way to each other but more efficiently than BM-MSCs (Figure 8).

## Figures and Tables

**Figure 1 ijms-25-09578-f001:**
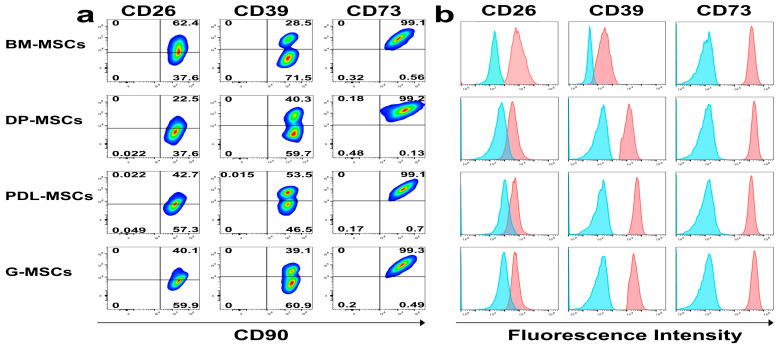
Expression of CD26, CD39, and CD73 in BM-MSCs, DP-MSCs, PDL-MSCs, and G-MSCs. The expression of CD26, CD39, and CD73 on the membrane was determined by means of flow cytometry, and the mean fluorescence intensity (MFI) is shown. (**a**) Dot plots representative of the percentage of MSCs expressing CD26, CD39, and CD73 (CD90 is shown as a reference molecule of MSCs population). (**b**) Representative histograms of the fluorescence intensity of CD26, CD39, and CD73 in MSCs. (**c**) Percentage of CD26-, CD39-, and CD73-expressing cells among the population of MSCs. (**d**) Increase in the number of CD26-, CD39-, and CD73-positive cells according to the MFI. The results represent the mean ± SD from three independent experiments. *n* = 9 replicates/source. * Significant difference compared with BM-MSCs (*p* < 0.05).

**Figure 2 ijms-25-09578-f002:**
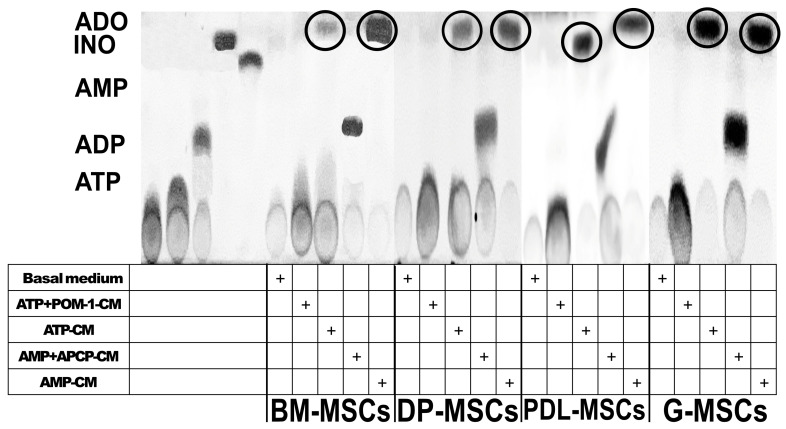
Enzymatic activity of CD39 and CD73 in BM-MSCs, DP-MSCs, PDL-MSCs, and G-MSCs. Representative image of ADO production from ATP or AMP, 2 × 10^6^ MSCs were cultured in serum-free medium supplemented with 5 mM ATP (ATP-CM) or AMP (AMP-CM) for 5 h; in some cultures, no substrate was added (basal medium), or 10 μM of the CD39 inhibitor (POM-1; ATP+POM-1-CM) or the CD73 inhibitor (APCP; AMP+APCP-CM) plus ATP or AMP, respectively, were added. The production of ADO was subsequently evaluated on TLC plates. From left to right, ATP, ADP, AMP, ADO, and INO standards were added, followed by the addition of MSC-CM with or without a substrate.

**Figure 3 ijms-25-09578-f003:**
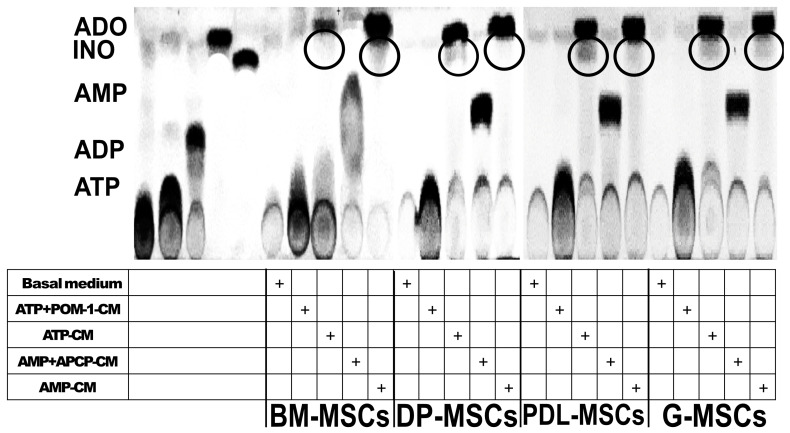
Production of INO by BM-MSCs, DP-MSCs, PDL-MSCs, and G-MSCs. Representative image of INO production from ATP or AMP. A total of 2 × 10^6^ MSCs were cultured in serum-free medium supplemented with 5 mM ATP (ATP-CM) or AMP (AMP-CM) for 24 h; in some cultures, no substrate was added (basal medium), or 10 μM of the CD39 inhibitor (POM-1; ATP+POM-1-CM) or the CD73 inhibitor (APCP; AMP+APCP-CM) plus ATP or AMP, respectively, were added. The production of INO was subsequently evaluated on TLC plates. From left to right, ATP, ADP, AMP, ADO, and INO standards were added, followed by the addition of MSC-CM with or without a substrate.

**Figure 4 ijms-25-09578-f004:**
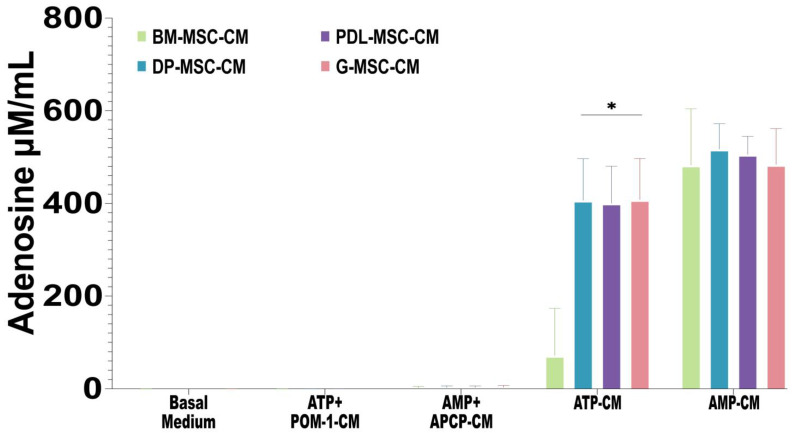
ADO concentrations in the BM-MSC-CM, DP-MSC-CM, PDL-MSC-CM, and G-MSC-CM. With respect to the concentration of ADO generated from ATP or AMP, 2 × 10^6^ MSCs were grown in serum-free medium supplemented with 5 mM ATP (ATP-CM) or AMP (AMP-CM) for 5 h; in some cultures, no substrate was added (basal medium), or 10 μM of the CD39 inhibitor (POM-1; ATP + POM-1-CM) or the CD73 inhibitor (APCP; AMP + APCP-CM) plus ATP or AMP, respectively, were added. The ADO concentration was evaluated via UPLC, and the concentrations were expressed in μM/mL. The results represent the mean ± SD from three independent experiments. *n* = 9 replicates/source. * Significant difference compared with BM-MSCs (*p* < 0.05).

**Figure 5 ijms-25-09578-f005:**
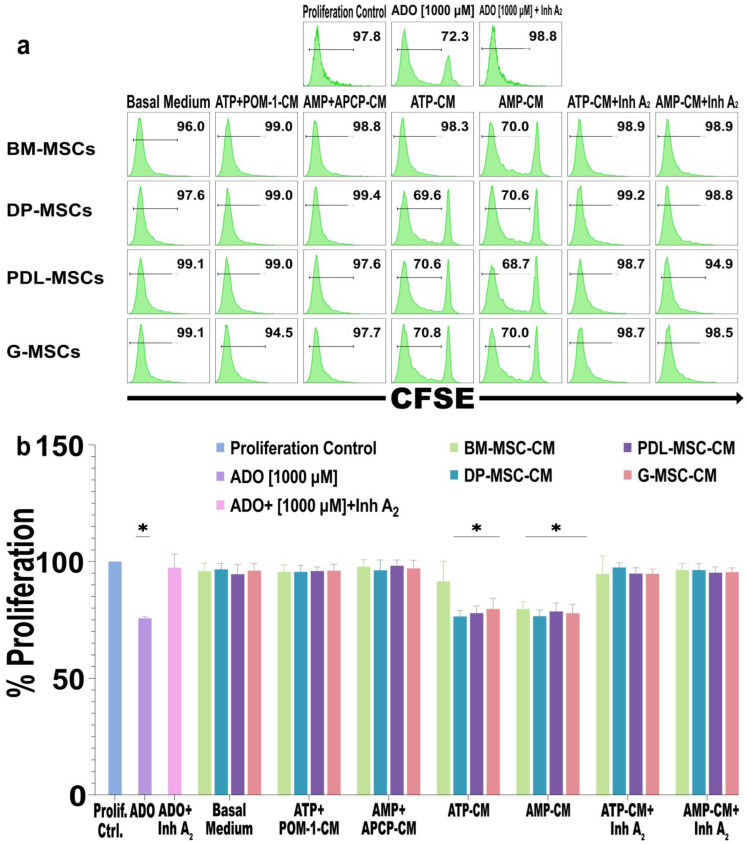
Effects of CM from BM-MSCs, DP-MSCs, PDL-MSCs, and G-MSCs on CD3^+^ T lymphocytes. A total of 1 × 10^5^ CD3^+^ lymphocytes were stained with CSFE, activated with CD2/CD3/CD28 beads, and cultured for 5 days with 20% CM from MSC cultures. The CM were obtained after 5 h of culture in the absence (basal medium) or presence of substrates (MSC-ATP-CM, MSC-AMP-CM) and inhibitors of ectoenzymes (ATP+POM-1-CM or AMP+APCP-CM). In some lymphocyte cultures, 2 μM of each specific inhibitor of the A_2_A and A_2_B receptors (ZM241385 and MRS1754, respectively; MSC-ATP-CM+Inh A_2_ or MSC-AMP-CM+Inh A_2_) were added. Lymphocyte proliferation was evaluated by means of flow cytometry. (**a**) Representative histogram of the percentage of CFSE-labeled CD3^+^ T cell proliferation. (**b**) Bar graph of the normalized percentage of proliferative CD3^+^ T cells. The results represent the mean ± SD from three independent experiments. *n* = 9 replicates/source. * Significant difference with respect to the proliferation control (*p* < 0.05).

**Figure 6 ijms-25-09578-f006:**
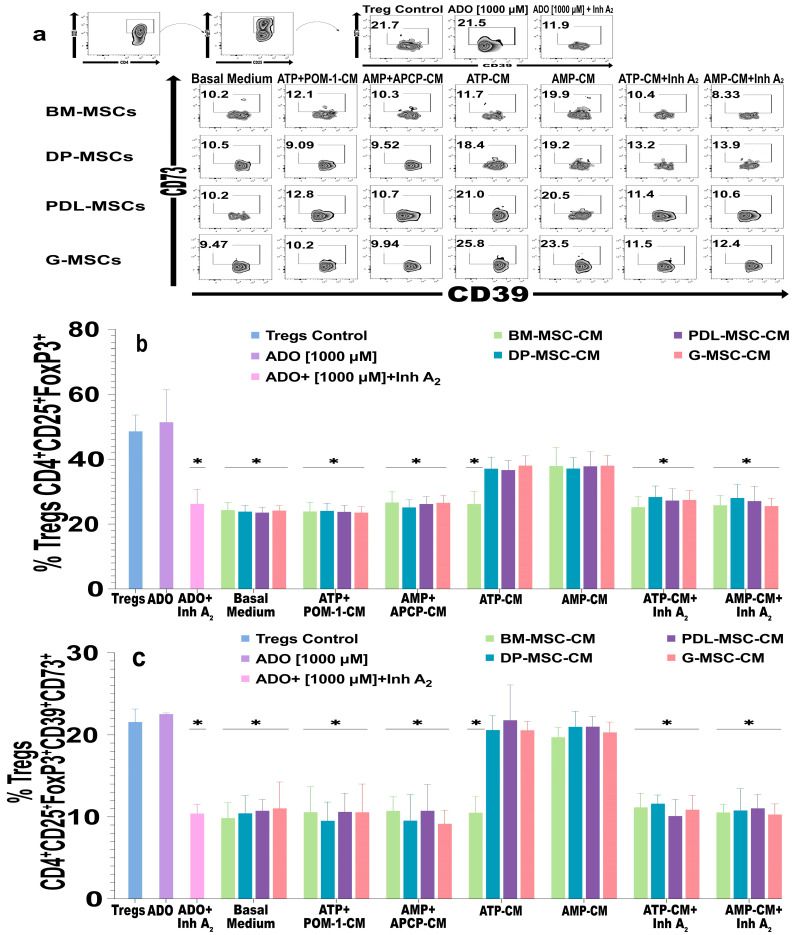
Effects of CM from BM-MSCs, DP-MSCs, PDL-MSCs, and G-MSCs on CD4^+^ T lymphocytes. A total of 1 × 10^5^ CD4^+^ lymphocytes were activated with CD2/CD3/CD28 beads and cultured for 4 days with 20% CM from MSC cultures. The CM were obtained after 5 h of culture in the absence (basal medium) or presence of substrates (MSC-ATP-CM, MSC-AMP-CM) and inhibitors of ectoenzymes (ATP+POM-1-CM or AMP+APCP-CM). In some lymphocyte cultures, 2 μM of each specific inhibitor of the A_2_A and A_2_B receptors (ZM 241385 and MR S1754, respectively; MSC-ATP-CM+Inh A_2_ or MSC-AMP-CM+Inh A_2_) were added. Lymphocyte differentiation was evaluated by means of flow cytometry. (**a**) Representative plot of the percentage of CD4^+^CD25^+^FoxP3^+^ CD39^+^CD73^+^ Treg lymphocytes. (**b**) Effect of MSC-CM on the differentiation of CD4^+^CD25^+^FoxP3^+^ lymphocytes. (**c**) Expression of CD39^+^CD73^+^ in Treg lymphocytes cultured in the presence of MSC-CM. The results represent the mean ± SD from three independent experiments. *n* = 9 replicates/source. * Significant difference with respect to the Treg control (*p* < 0.05).

**Figure 7 ijms-25-09578-f007:**
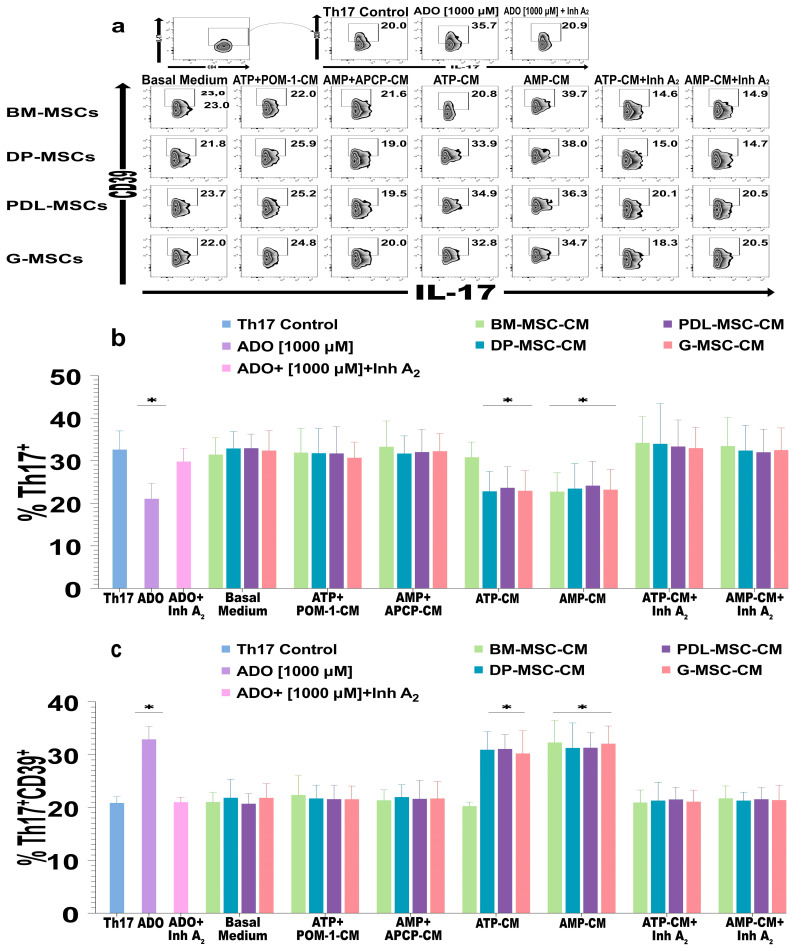
Effects of CM from cultures of BM-MSCs, DP-MSCs, PDL-MSCs, and G-MSCs on the generation of Th17 lymphocytes. A total of 1 × 10^5^ CD4^+^ lymphocytes activated with CD2/CD3/CD28 beads were differentiated with IL-1β + IL-6 + IL-23 + TGF-β1 and cultured for 4 days with 20% CM from MSC cultures. The CM were obtained after 5 h of culture in the absence (basal medium) or presence of substrates (MSC-ATP-CM, MSC-AMP-CM) and inhibitors of ectoenzymes (ATP+POM-1-CM or AMP+APCP-CM). In some lymphocyte cultures, 2 μM of each specific inhibitor of the A_2_A and A_2_B receptors (ZM 241385 and MRS 1754, respectively; MSC-ATP-CM+Inh A_2_ or MSC-AMP-CM+Inh A_2_) were added. The effect on CD4^+^ lymphocytes was evaluated via flow cytometry. (**a**) Plot representative of the percentage of Th17^+^CD39^+^ lymphocytes. (**b**) Effect of CM on the differentiation of Th17 lymphocytes. (**c**) Expression of CD39 in Th17 lymphocytes generated in the presence of MSC-CM. The results represent the mean ± SD from three independent experiments. *n* = 9 replicates/source. * Significant difference with respect to the Th17 control (*p* < 0.05).

**Figure 8 ijms-25-09578-f008:**
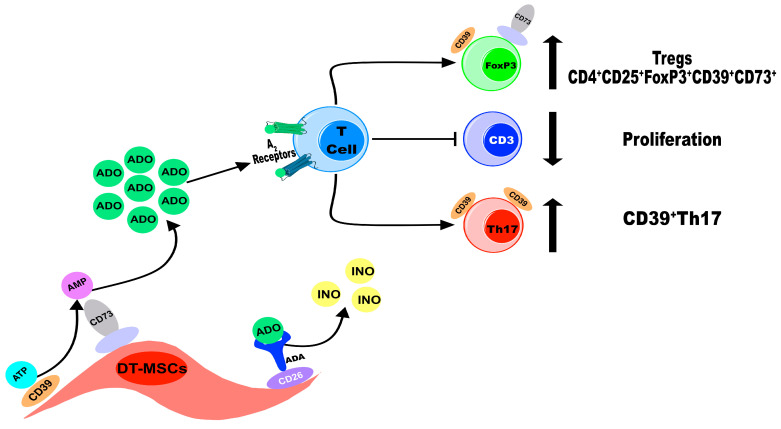
DT-MSCs use the adenosinergic pathway as an immunomodulatory mechanism. Schematic diagram of ADO production by DT-MSCs. The expression of the CD39 ectonueclotidase allows the dephosphorylation of extracellular ATP to AMP, which is transformed into ADO by CD73. The concentration of ADO generated from ATP decreases the proliferation of CD3^+^ T lymphocytes and induces the differentiation of CD4^+^CD25^+^FoxP3^+^ Tregs that coexpress CD39^+^ and CD73^+^. Additionally, the concentration of ADO generated from ATP by DT-MSCs also induces the expression of CD39 in Th17 lymphocytes. Finally, DT-MSCs express CD26 and produce low concentrations of INO, which suggests that they express ADA to regulate ADO concentrations.

## Data Availability

The original contributions presented in this study are included in the article; further inquiries can be directed to the corresponding author.

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
