# Peer review of "Mesenchymal Stem/Stromal Cells Derived from Dental Tissues Mediate the Immunoregulation of T Cells through the Purinergic Pathway"

_ijms, 2024, doi:10.3390/ijms25179578_

Round 1

Reviewer 1 Report

Comments and Suggestions for Authors

In this article, the authors discovered that human dental tissue mesenchymal stem cells (DT-MSCs) exhibit a greater capacity to generate ADO from ATP compared to bone marrow-derived mesenchymal stem cells (BM-MSCs), thereby demonstrating an adenosinergic immunomodulatory property. However, there are several concerns regarding the study design and results.

1. The rationale for selecting the adenosinergic pathway among different types of stem cells should be addressed through unbiased experiments, such as RNAseq analysis, to confirm whether CD26, CD39, CD73, and the adenosinergic pathway are indeed significantly differentially expressed pathways.

2. To accurately analyze cell metabolism, Seahorse experiments should be conducted to quantify real-time ATP production.

3. In order to support the in vitro observations made in this study, further in vivo animal experiments investigating the anti-inflammatory properties of DT-MSCs compared with BM-MSCs are necessary.

4. Since it is mentioned that BM-MSCs possess regenerative and immunomodulatory properties and are utilized in clinical treatments for various diseases, it is important to investigate the regenerative potential of DT-MSCs as alternative sources to BM-MSCs.

5. Figure 2 shows that BM-MSCs produce more ADO from AMP than DT-MSCS; however, these results contradict those presented in figure 4.

6. Several figures have low quality and appear distorted. The blots from TLC plates were unclear which raises uncertainty about their validity. Additionally, some figures appear identical or repetitive; particularly noticeable is the flow cytometry analysis of CD14 and HLA-DR. Furthermore, Figure 1c lacks labels on its X-axis which should be revised by the authors.

Comments on the Quality of English Language

Minor changes of the English Language.

Reviewer 2 Report

Comments and Suggestions for Authors

We had carefully read the manuscipt entitled: “Mesenchymal Stem/Stromal Cells Derived from Dental Tissues Mediate Immunoregulation of T Cells through the Purinergic Pathway” from Dr Luis Ignacio Poblano-Pérez et al. submitted to International Journal of Molecular Sciences.

The authors want to evaluate the potential role of the purinergic pathway for immunomodulation in the case of the use of mesemchymal stem cells (MSC) derived from dental tissues (pulp, periodontal ligament or gingival tissue). They used BM-MSC (one of the most cited MSC) as control. After demonstrating the stemness and identity of their MSCs, the authors show the cells express the molecules related to the purinergic pathway and demonstrate the pathway is active with production of adenosine (ADO) and its degradation in inosine (INO). Then they use conditioned media to evaluate the modulation of T cells.

There is some minor points which can be corrected.

ADO should be written in full name in the abstract.

Line 53 , replace exploit with utilize.

Line 235, “ p0atients” typo error.

Line 552, crops? Did you mean groups?

Comments:

Major comments

The age of the donors, their health conditions, and other factors should be indicated in the method section, as these parameters affect the immunomodulatory properties of the MSCs.

The authors discussed the potential applicability of DT-MSCs as an alternative option for BM-MSCs. the fact that DT-MSCs scarcity necessitates their in vitro expansion should be highlighted. This could change their immunomodulatory activity.

It can be interesting to cocultivate MSCs and immune cells in a coculture experiment. Also, including macrophages in the study and evaluating the effects of ADO on these cells would further strengthen a potential clinical translation of this research.

Minor comments

Fig.2 and 3: graphs with quantification are necessary (with stats). Some of them (but not all) are in Fig.4 but the data should be with the images.

In figure 4, the concentrations of ADO in G-MSCs was significantly lower than DP-MSCs. However, in figure 5, the decrease in lymphocytes proliferation was similar between G-MSCs and DP-MSCs groups. There is a tendency for oversimplification of the study design. There must be other signaling molecules and factors that might play a role in the effects of MSCs-derived conditioned media on the lymphocyte’s behavior.

In page 9, most of the text should be part of the introduction, not the results.

Units in figures should be consistent (µM or µg/ml). It is important to be able to compare the data in Fig.4 and subsequent. (1000 µM ADO is used as ref but secretion is around 7 µg/ml… difficult to know if the control is appropriate (we have to calculate J).

The police and size of the text in the figures should be standardized.

Fig.8 should be rethink to be more informative. At the present format it is like a “do it yourself”.
